# The role of anterior insular cortex inputs to dorsolateral striatum in binge alcohol drinking

**David L Haggerty[1], Braulio Munoz[1], Taylor Pennington[1], Gonzalo Viana Di Prisco[1], Gregory G Grecco[1,2], Brady K Atwood[1,3]\***

[1]Department of Pharmacology & Toxicology, Indiana University School of Medicine, Indianapolis, United States; [2]Medical Scientist Training Program, Indiana University School of Medicine, Indianapolis, United States; [3]Stark Neurosciences Research Institute, Indiana University School of Medicine, Indianapolis, United States

**Abstract** How does binge drinking alcohol change synaptic function, and do these changes maintain binge consumption? The anterior insular cortex (AIC) and dorsolateral striatum (DLS) are brain regions implicated in alcohol use disorder. In male, but not female mice, we found that binge drinking alcohol produced glutamatergic synaptic adaptations selective to AIC inputs within the DLS. Photoexciting AIC→DLS circuitry in male mice during binge drinking decreased alcohol, but not water consumption and altered alcohol drinking mechanics. Further, drinking mechanics alone from drinking session data predicted alcohol-related circuit changes. AIC→DLS manipulation did not alter operant, valence, or anxiety-related behaviors. These findings suggest that alcohol-mediated changes at AIC inputs govern behavioral sequences that maintain binge drinking and may serve as a circuit-based biomarker for the development of alcohol use disorder.

**\*For correspondence:**
bkatwood@iu.edu

**Competing interest:** The authors declare that no competing interests exist.

## Editor's evaluation

Haggerty et al. reported findings examining how changes in brain function are involved in alcohol binge drinking, with a selective focus on the synaptic and circuit alterations that occur in the anterior insular cortex (AIC) inputs within the dorsolateral striatum (DLS). They show that chronic alcohol drinking produces glutamatergic synaptic adaptations in male mice and by stimulating this circuit binge drinking could be reduced without altering either water consumption or general performance for select reinforcing, anxiogenic or locomotor behaviors. The results of this study may specifically improve our understanding of the sex-specific differences in neurocircuitry mediating excessive drinking associated with alcohol use disorder.

## Introduction

Binge alcohol consumption, defined as consuming at least four drinks for women and five drinks for men in a 2 hr drinking session, represents a large proportion of the deaths associated with problematic alcohol use and alcohol use disorder (AUD) (*Mokdad et al., 2004*; *Kanny et al., 2018*). Binge drinking is particularly prevalent among young adults, is behaviorally conserved in rodents, and is a theoretical entry point to the addiction cycle (*Thiele and Navarro, 2014*; *Patrick et al., 2013*; *Chung et al., 2018*; *Koob and Volkow, 2010*). Ultimately, partaking in binge drinking is one of the strongest predictors for developing AUD (*Addolorato et al., 2018*).

How alcohol alters neural circuitry that underlies binge drinking remains poorly understood. Previous work has shown binge drinking alters glutamate receptor function across many brain regions

(*Hwa et al., 2017*). Few neural circuits, such as the ventral tegmental area and thalamic inputs to the extended amygdala, locus coeruleus outputs to lateral hypothalamus and rostromedial tegmental nucleus, and medial prefrontal cortex inputs to the periaqueductal gray have been directly linked to the behavioral control of binge alcohol drinking (*Rinker et al., 2017*; *Ferguson et al., 2019*; *Levine et al., 2021*; *Dornellas et al., 2021*; *Burnham et al., 2021*; *Siciliano et al., 2019*).

Numerous clinical and preclinical studies directly implicate a role for the insular cortex in encoding responses to alcohol cues and consumption (*Centanni et al., 2021*; *Gogolla, 2017*). Specifically in rodents, anterior insular cortex (AIC) inputs to nucleus accumbens core have been shown to drive aversion-resistant alcohol consumption and play a role in alcohol discrimination and self-administration (*Jaramillo et al., 2018a*; *Jaramillo et al., 2018b*; *Seif et al., 2013*). Others have shown that principal AIC neurons can govern fluid consumption across thirst states that influence voluntary consumption of rewarding substances, such as alcohol (*Haaranen et al., 2020*; *Zhao et al., 2020*). Yet, AIC neurons send inputs to many downstream brain regions, so resolving how alcohol alters the functional connectivity by input specificity is of great importance (*Gehrlach et al., 2020*). Investigating how the AIC governs behavioral control over alcohol consumption across drinking paradigms and brain states at various input locations is essential to advance our understanding of the neuroadaptations associated with AUD.

Recent studies have demonstrated a direct anatomical projection from the AIC to the dorsolateral striatum (DLS) (*Hunnicutt et al., 2016*; *Muñoz et al., 2018*; *Muñoz et al., 2020*; *Wall et al., 2013*). The DLS is a region of striatum that is implicated in AUD and behavioral responding for alcohol (*Barker et al., 2015*; *Campbell and Lawrence, 2021*). Glutamate signaling within the DLS also mediates alcohol seeking behaviors and alcohol exposure alters excitatory glutamate transmission and synaptic plasticity within the DLS (*Corbit et al., 2014*; *Muñoz et al., 2018*; *Abburi et al., 2016*; *Johnson et al., 2020*; *Rangel-Barajas et al., 2021*; *DePoy et al., 2013*). We also previously showed that glutamatergic synaptic plasticity at AIC inputs to the DLS (AIC→DLS) is uniquely vulnerable to disruption by alcohol exposure, relative to other glutamate synapses within the DLS (*Muñoz et al., 2018*).

Thus, we sought to explore in greater depth the impact that binge alcohol consumption has on glutamatergic transmission and synaptic plasticity specifically at these AIC inputs to the DLS. While these synapses have been investigated anatomically and functionally, they have yet to be behaviorally evaluated in any context. Therefore, we also sought to determine how these synapses may regulate binge alcohol consumption.

## Results and discussion

We isolated AIC→DLS circuitry by injecting an anterograde adeno-associated virus (AAV) into the AIC, utilizing the *CaMKIIa* promoter to drive expression of channelrhodopsin-2 (ChR2) tagged with enhanced yellow fluorescent protein (EYFP) to visualize and modulate glutamatergic AIC inputs within the DLS (*Figure 1A* and *Figure 1—figure supplement 1*). Animals then underwent 3 weeks of the Drinking in the Dark (DID) paradigm where they had 0, 2, or 4 hr of access to alcohol or water each day, which produced 'binge-like' levels of alcohol intake as measured by a positive correlation between alcohol intake and blood alcohol concentration (BAC) with a subset of animals achieving BACs greater than 0.08 mg% (*Figure 1B–D* and *Figure 1—figure supplement 2*; *Thiele and Navarro, 2014*). Twenty-four hours after the final DID session, once 3 weeks of DID was completed, acute slices of the DLS were made to measure AIC-mediated synaptic responses.

An input-output assessment of optically evoked excitatory postsynaptic currents (oEPSCs) recorded from DLS medium spiny neurons (MSNs) at increasing light intensities showed that alcohol drinking animals had greater oEPSC amplitudes than water controls (*Figure 1E*). Binge drinking alcohol also decreased optically evoked AMPA to NMDA glutamate receptor current ratios (oAMPA /oNMDA) (*Figure 1F*). Given that the increased oEPSCs measured with our input-output assessment (*Figure 1E*) were likely AMPA receptor-mediated, we interpret the decrease in oAMPA /oNMDA ratio to mean that alcohol consumption has a larger effect on NMDA receptor currents than AMPA receptor currents. Although, assessing paired-pulse ratios of oEPSCs (oPPR) at increasing interstimulus intervals showed no significant change (*Figure 1G*). Binge alcohol consumption also slightly decreased amplitudes of spontaneous excitatory postsynaptic currents (sEPSCs), but there were no other differences

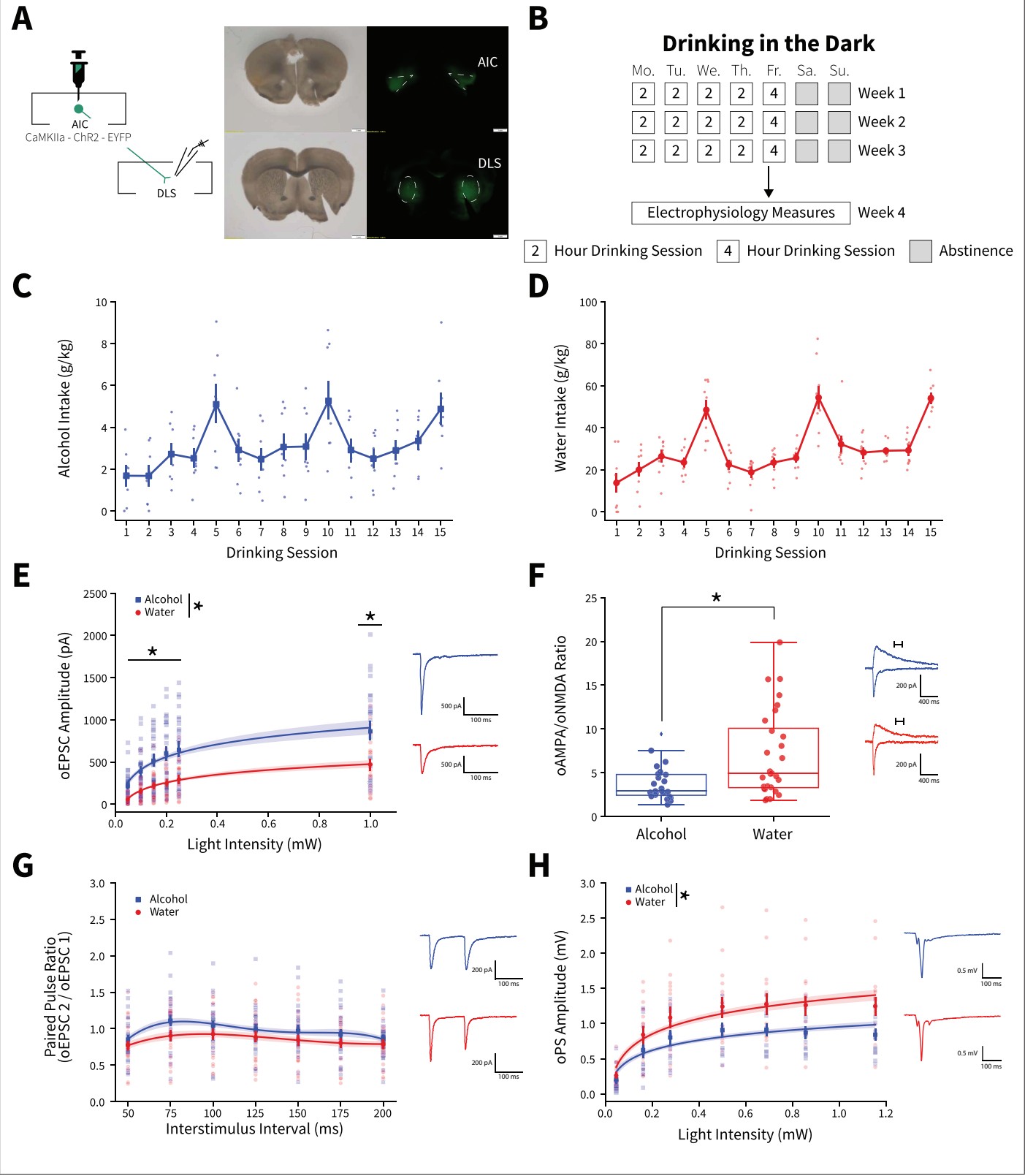

**Figure 1.** Binge alcohol consumption alters synaptic plasticity of anterior insular cortex (AIC) inputs to the dorsolateral striatum (DLS). (**A**) Representation of viral strategy and expression to record AIC input responses within the DLS. (**B**) Schematic of Drinking in the Dark (DID) protocol for electrophysiology experiments. (**C**) Group and individual animals' alcohol consumption (n=8 animals). (**D**) Group and individual animals' water consumption (n=10 animals). (**E**) Binge alcohol consumption increased optically evoked postsynaptic current (oEPSC) amplitude of AIC inputs within

*Figure 1 continued on next page*

*Figure 1 continued*

the DLS (two-way mixed analysis of variance (ANOVA), Fluid $F_{(1,50)}$ = 15.6084, p=0.0002; Fluid × Light Intensity $F_{(5,250)}$ = 6.0674, p=0.000025; alcohol: 26 recordings, n=5 animals; water: 26 recordings, n=6 animals). (**F**) Binge alcohol consumption reduced optically evoked AMPA (oAMPA) to NMDA (oNMDA) glutamate receptor current ratios (Mann-Whitney, *U*=189, p=0.0072; alcohol: 24 recordings, n=5 animals; water: 28 recordings, n=6 animals). (**G**) There was no main effect of binge alcohol consumption on paired-pulse ratios of oEPSCs (oPPR) compared to water consumption (two-way mixed ANOVA, Fluid $F_{(1,55)}$ = 2.5544, p=0.1157; alcohol: 30 recordings, n=6 animals; water: 27 recordings, n=5 animals). (**H**) Binge drinking alcohol decreased optically evoked population spike (oPS) amplitudes produced by photoexciting AIC inputs within the DLS (two-way mixed ANOVA, Fluid $F_{(1,28)}$ = 4.3484, p=0.0463; Fluid × Light Intensity $F_{(6,168)}$ = 2.9574, p=0.0090; alcohol: 14 recordings, n=4 animals; water: 16 recordings, n=6 animals). Error bars and shading indicate ± SEM. Box plot whiskers represent interquartile range.

The online version of this article includes the following source data and figure supplement(s) for figure 1:

**Source data 1.** Binge alcohol consumption alters synaptic plasticity of anterior insular cortex (AIC) inputs to the dorsolateral striatum (DLS).

**Figure supplement 1.** Locations of anterior insular cortex (AIC) injections for dorsolateral striatum (DLS) electrophysiology.

**Figure supplement 2.** Binge alcohol consumption is positively correlated with blood alcohol concentration.

**Figure supplement 2—source data 1.** Binge alcohol consumption is positively correlated with blood alcohol concentration.

**Figure supplement 3.** Binge alcohol consumption alters amplitudes of spontaneous excitatory synaptic activity in the dorsolateral striatum (DLS).

**Figure supplement 3—source data 1.** Binge alcohol consumption alters amplitudes of spontaneous excitatory synaptic activity in the dorsolateral striatum (DLS).

**Figure supplement 4.** Female binge alcohol consumption does not alter synaptic plasticity of anterior insular cortex (AIC) inputs to the dorsolateral striatum (DLS).

**Figure supplement 4—source data 1.** Female binge alcohol consumption does not alter synaptic plasticity of anterior insular cortex (AIC) inputs to the dorsolateral striatum (DLS).

**Figure supplement 5.** A single binge alcohol session alters optically evoked glutamate currents, but does not change the AMPA/NMDA current ratio or alter dorsolateral striatum (DLS) network effects.

**Figure supplement 5—source data 1.** A single binge alcohol session alters optically evoked glutamate currents, but does not change the AMPA/NMDA current ratio or alter dorsolateral striatum (DLS) network effects.

in spontaneous synaptic activity between water and alcohol drinkers in the DLS (*Figure 1—figure supplement 3*).

Consistent with previous findings, we have shown that alcohol can have synapse-specific effects in dorsal striatum, which are otherwise obscured by the net effects at other synapses when synaptic transmission is measured in a non-circuit-specific manner (*Muñoz et al., 2018*; *Muñoz et al., 2020*). These data suggest that the measured alcohol-mediated plasticity changes were selective for AIC inputs. Specifically, these data indicate that binge drinking alcohol selectively potentiates postsynaptic AMPA and NMDA glutamate receptor responses at AIC→DLS synapses. We acknowledge that we cannot exclude that alcohol produces other distinct changes at non-AIC inputs to DLS.

To determine if the measured alterations in synapse-specific binge alcohol consumption were conserved in female mice, we exposed aged-matched female mice to 3 weeks of DID after virally isolating AIC→DLS circuitry to perform electrophysiology experiments that methodologically replicated our male recordings (*Figure 1—figure supplement 4A-C*). Interestingly, female mice that binge drank alcohol compared to water drinking controls show no changes in oEPSC amplitude, oAMPA/oNMDA ratio, or oPPR (*Figure 1—figure supplement 4D-F*). With both males and female datasets combined, we found a significant fluid effect, but not a significant fluid by sex effect for oEPSC amplitude (three-way mixed analysis of variance [ANOVA], Fluid $F_{(1,75)}$ = 8.309, p=0.0051; Fluid × Sex $F_{(1,75)}$ = 3.044, p=0.0851). For oAMPA/oNMDA ratios, we discovered a sex effect driven by binge alcohol exposed male mice (two-way ANOVA, Sex $F_{(1,80)}$ = 5.8532, p=0.0178; Fluid × Sex $F_{(1,80)}$ = 7.2132, p=0.0088). Since the observed synaptic plasticity changes measured were present only in male mice, we chose to focus on males for further AIC focused circuit-specific and behavioral analyses, but other female circuit-specific alterations need to be explored in future studies.

To investigate the effects of activating AIC inputs on broader DLS network activity, we recorded optically evoked population spikes (oPSs) produced by photoexciting AIC inputs within the DLS at increasing light intensities. oPS amplitudes were decreased in male mice that binge drank alcohol compared to water (*Figure 1H*), indicating that the net effect of binge alcohol-induced AIC→DLS synaptic adaptations is a reduced network response to AIC input activation. This is curious as we measured an increased glutamate receptor response (*Figure 1E and F*).

Furthermore, we tested the possibility that only a single DID session was necessary to induce changes in oEPSCs, synaptic plasticity, and altered DLS network effects (*Figure 1—figure supplement 5A-D*). A single alcohol DID exposure increased oEPSC amplitude compared to water controls, a finding consistent with 3 weeks of DID exposure (*Figure 1—figure supplement 5E*). Yet, there were no changes in oAMPA/oNDMA ratio and oPS amplitudes (*Figure 1—figure supplement 5F and G*). This suggests that a single DID exposure is not sufficient to recapitulate the AIC→DLS circuit changes seen after 3 weeks of DID exposure.

Together, these data indicate that enhanced glutamate transmission at AIC inputs to DLS may drive local inhibitory networks leading to overall decreased MSN activity. AIC inputs equally innervate different types of MSNs indicating that this effect is not due to a preferential engagement of one type of MSN (*Wall et al., 2013*). Other possibilities are that AIC inputs induce feedforward inhibition through local interneurons, such as engaging fast-spiking interneurons, which have been implicated in compulsive alcohol consumption (*Patton et al., 2021*; *Manz et al., 2020*). This is a possibility that will need to be explored in future studies.

Alterations in DLS neurotransmission are known to facilitate the development of habit learning in the context of substance-related behaviors and are also associated with the expression of numerous alcohol-related behaviors (*Koob and Volkow, 2010*; *Lovinger and Alvarez, 2017*; *Corbit et al., 2012*; *Patton et al., 2016*). Glutamate receptor signaling specifically within the DLS mediates alcohol seeking behavior (*Corbit et al., 2014*). We hypothesized that by modulating AIC inputs within the DLS, we could alter ongoing binge drinking behavior. Initially, we predicted that photoexciting AIC inputs during alcohol consumption would increase binge drinking.

We again isolated AIC→DLS circuitry by injecting an AAV-ChR2-EYFP vector or a control vector that solely expressed enhanced green fluorescent protein (eGFP) into the AIC and implanted a wireless, unilateral optogenetic probe into the DLS to modulate AIC inputs to the DLS during homecage DID sessions (*Figure 2A* and *Figure 2—figure supplement 1*; *Shin et al., 2017*). Animals then underwent 3 weeks of water or alcohol DID, via lickometers that monitored liquid intake, to acquire binge alcohol-induced plasticity changes in the absence of optical stimulation (*Figure 2B–C*; *Godynyuk et al., 2019*). Licks, bouts, and total lick duration metrics from the lickometers were significantly correlated with water and alcohol intakes and thus provided high-resolution drinking microstructure details about how and when during each DID session each animal consumed water or alcohol (*Figure 2—figure supplement 2*). During binge acquisition (weeks 1–3), there were baseline differences between ChR2 and eGFP controls for water, but not alcohol intake (*Figure 2D–E*).

Following binge acquisition, animals underwent another 3 weeks of water or alcohol DID. In one half of the animals the detection of an alcohol or water lick triggered the activation of the unilateral optogenetic probe in a closed-loop manner, delivering blue light (470 nm) to both ChR2 and eGFP controls for the duration of their licking behavior to evoke glutamate release from AIC inputs to the DLS or as a blue light control, respectively (*Figure 2F*). The other half of the animals received blue light stimulation stochastically throughout each DID session (see Materials and methods), but there were no differences in water or alcohol intakes between closed-loop and open-loop light delivery, so we collapsed on these groupings (*Figure 2—figure supplement 3*).

To account for baseline differences between viral expression within fluid type, we summed intakes and microstructure features by week within each animal and displayed these intakes and microstructure features as a percent change from week 3 to determine how photoexciting AIC inputs within the DLS altered intakes and microstructure features. For alcohol and water intake measures for all individual sessions, see *Figure 2—figure supplement 4*.

For binge evaluation (weeks 4–6), there was no change in water intake between ChR2 and eGFP controls (*Figure 2G*). Yet, contrary to our initial hypothesis, ChR2 expressing animals drank significantly less alcohol than eGFP controls. Thus, driving glutamate release from AIC inputs within the DLS selectively decreased binge drinking in animals that consumed alcohol, but not water (*Figure 2H*). This was not a product of the in vivo optical stimulation inducing differential plasticity in water and alcohol drinkers. Testing the same stimulation pattern used in vivo in brain slices did not produce different effects in slices from water and alcohol drinkers or induce long-lasting glutamatergic plasticity on its own, even in the presence of the GABAA receptor antagonist picrotoxin (*Figure 2—figure supplement 5*).

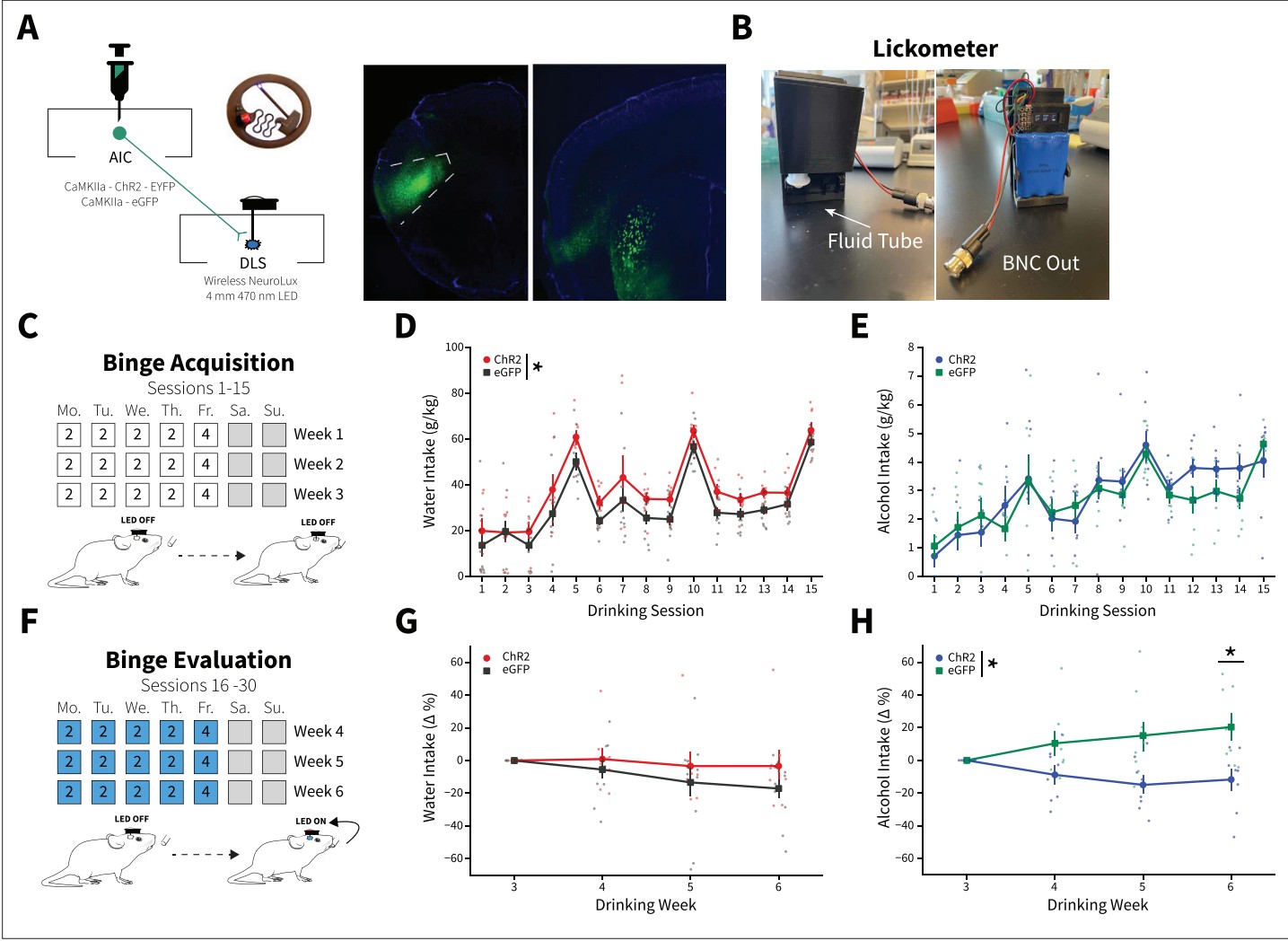

**Figure 2.** Anterior insular cortex (AIC) inputs to the dorsolateral striatum (DLS) modulate binge alcohol intake, but not water intake. (**A**) Representation of viral strategy and expression to modulate AIC inputs within the DLS. (**B**) Photographs and description of lickometer. (**C**) Schematic of Drinking in the Dark (DID) protocol for binge alcohol or water acquisition. (**D**) Group and individual animals' water consumption differed by viral expression during binge acquisition (two-way mixed analysis of variance [ANOVA], Virus $F_{(1,16)}$ = 8.1951, p=0.0113; ChR2: n=8 animals, eGFP: n=11 animals). (**E**) Group and individual animals' alcohol consumption did not differ by viral expression grouping during binge acquisition (two-way mixed ANOVA, Virus $F_{(1,14)}$ = 0.1829, p=0.6754; ChR2: n=8 animals, eGFP: n=8 animals). (**F**) Schematic of DID protocol for binge alcohol or water evaluation. (**G**) Photoexciting AIC inputs within the DLS during DID did not alter water intakes (two-way mixed ANOVA, Virus $F_{(1,17)}$ = 1.0087, p=0.3293; ChR2: n=8 animals, eGFP: n=11 animals), but (**H**) decreased alcohol intake (two-way mixed ANOVA, Virus $F_{(1,14)}$ = 8.5743, p=0.0110; Virus × Drinking Week $F_{(3,42)}$ = 4.7132, p=0.0063; Drinking Week 6, p=0.0281; ChR2: n=8 animals, eGFP: n=8 animals). Error bars indicate± SEM. All post hoc comparisons are Sidak corrected.

The online version of this article includes the following source data and figure supplement(s) for figure 2:

**Source data 1.** Anterior insular cortex (AIC) inputs to the dorsolateral striatum (DLS) modulate binge alcohol intake, but not water intake.

**Figure supplement 1.** Locations of anterior insular cortex (AIC) injections and dorsolateral striatum (DLS) wireless optogenetic probe placements.

**Figure supplement 2.** Microstructure features measured from lickometers correlate with alcohol and water intakes.

**Figure supplement 2—source data 1.** Microstructure features measured from lickometers correlate with alcohol and water intakes.

**Figure supplement 3.** Channelrhodopsin-2 (ChR2) expressing yoked animals do not differ from closed-loop stimulated animals in alcohol and water intakes.

**Figure supplement 3—source data 1.** Channelrhodopsin-2 (ChR2) expressing yoked animals do not differ from closed-loop stimulated animals in alcohol and water intakes.

**Figure supplement 4.** Water and alcohol intakes for all Drinking in the Dark (DID) sessions.

**Figure supplement 4—source data 1.** Water and alcohol intakes for all Drinking in the Dark (DID) sessions.

*Figure 2 continued on next page*

*Figure 2 continued*

**Figure supplement 5.** Effect of using in vivo optogenetic stimulation parameters in brain slices.

**Figure supplement 5—source data 1.** Effect of using in vivo optogenetic stimulation parameters in brain slices.

**Figure supplement 6.** Photoexciting anterior insular cortex (AIC) inputs to the dorsolateral striatum (DLS) did not alter alcohol intakes during binge acquisition.

**Figure supplement 6—source data 1.** Photoexciting anterior insular cortex (AIC) inputs to the dorsolateral striatum (DLS) did not alter alcohol intakes during binge acquisition.

We reasoned that since driving glutamate release during water DID sessions did not alter intakes, that alcohol-induced plasticity changes at AIC inputs to DLS were a prerequisite for decreasing alcohol intake. To confirm, we photoexcited AIC inputs to the DLS in a separate cohort of animals during binge acquisition (sessions 1–15) to see if animals would decrease their intakes or not acquire stable alcohol intakes. These animals did not show any differences between non-photoexcited alcohol drinkers suggesting alcohol-induced AIC→DLS plasticity changes are necessary to alter binge alcohol intakes (*Figure 2—figure supplement 6*). These data are consistent with our electrophysiology data showing that a single DID session did not produce equivalent synaptic changes as 3 weeks of DID (*Figure 1* and *Figure 1—figure supplement 5*).

To quantify behavioral alterations associated with fluid intakes, we analyzed drinking microstructure features across drinking weeks by fluid. For water drinkers, there were no differences between ChR2 and eGFP controls in licks, lick duration, bouts, latency to drink, mean inter-drink interval, or bouts in the first 30 min of each DID session (*Figure 3A–F*). As expected for alcohol drinkers, there were significant decreases in licks, lick duration, and bouts in ChR2 compared to eGFP controls (*Figure 3G–I*). For alcohol latency to drink and mean inter-drink-interval, there were no significant effects between groups (*Figure 3J and K*), but the number of alcohol bouts in the first 30 min (a measurement of 'front-loading' behavior) decreased for ChR2 compared to eGFP controls (*Figure 3L*; *Wilcox et al., 2014*).

Although we found significant changes in microstructure features between viral groupings for alcohol drinkers, there were group trends, variability between animals, and many more microstructure features that were not captured in our weekly analyses to characterize intake changes. To more robustly model microstructure changes, we constructed a feedforward artificial neural network using all 18 microstructure features (see *Figure 3—source code 1*) from each DID session for all animals to predictively classify both fluid (water vs. alcohol) and virus (ChR2 vs. eGFP) (*Figure 3M*; *Emmert-Streib et al., 2020*). After splitting our dataset into training and testing sets, performing sixfold stratified cross-validation (to ensure each training fold had a representative proportion of fluid type and viral expression samples), and training the network we achieved an average of 66.13% accuracy (2.6 times better than chance accuracy) on data previously unseen by the model to predict fluid type and viral expression from a single DID session solely from microstructure data (*Figure 3N–O*). Thus, AIC inputs within the DLS govern binge alcohol-related behaviors so strongly that we can reliably predict which animals received specific experimental manipulations based on how they consume fluid from a DID session.

Altogether, our drinking data indicate that alcohol-induced neuroplasticity at AIC inputs to DLS may serve as a gain-of-function that allows these synapses to maintain alcohol consumption. In the absence of alcohol, these synapses seem to play no role in modulating intake behaviors, likely due to the absence of a specific type of plasticity required to engage or bring these synapses 'online'. Once alcohol-induced plasticity at these synapses occurs, it likely enables the governance of alcohol drinking behaviors via complex control over the patterns of licking behavior. Future work is needed to determine if consumption of other rewarding substances causes plasticity at AIC inputs to similarly produce such a gain-of-function. We note that in our previous work binge-like sucrose consumption did not produce the same effect on AIC-DLS synaptic plasticity as alcohol (*Muñoz et al., 2018*).

We next questioned if the observed decreases in alcohol intake following photoactivation were a product of AIC inputs simply altering behavior in general. In one cohort of animals after the final week of DID, we performed real-time place preference (RTPP) (*Figure 4A*). There were no differences between ChR2 and eGFP controls for time in zone or distance traveled, suggesting photoexciting AIC inputs to the DLS does not alter preference/avoidance behaviors or locomotion (*Figure 4B and C*). In

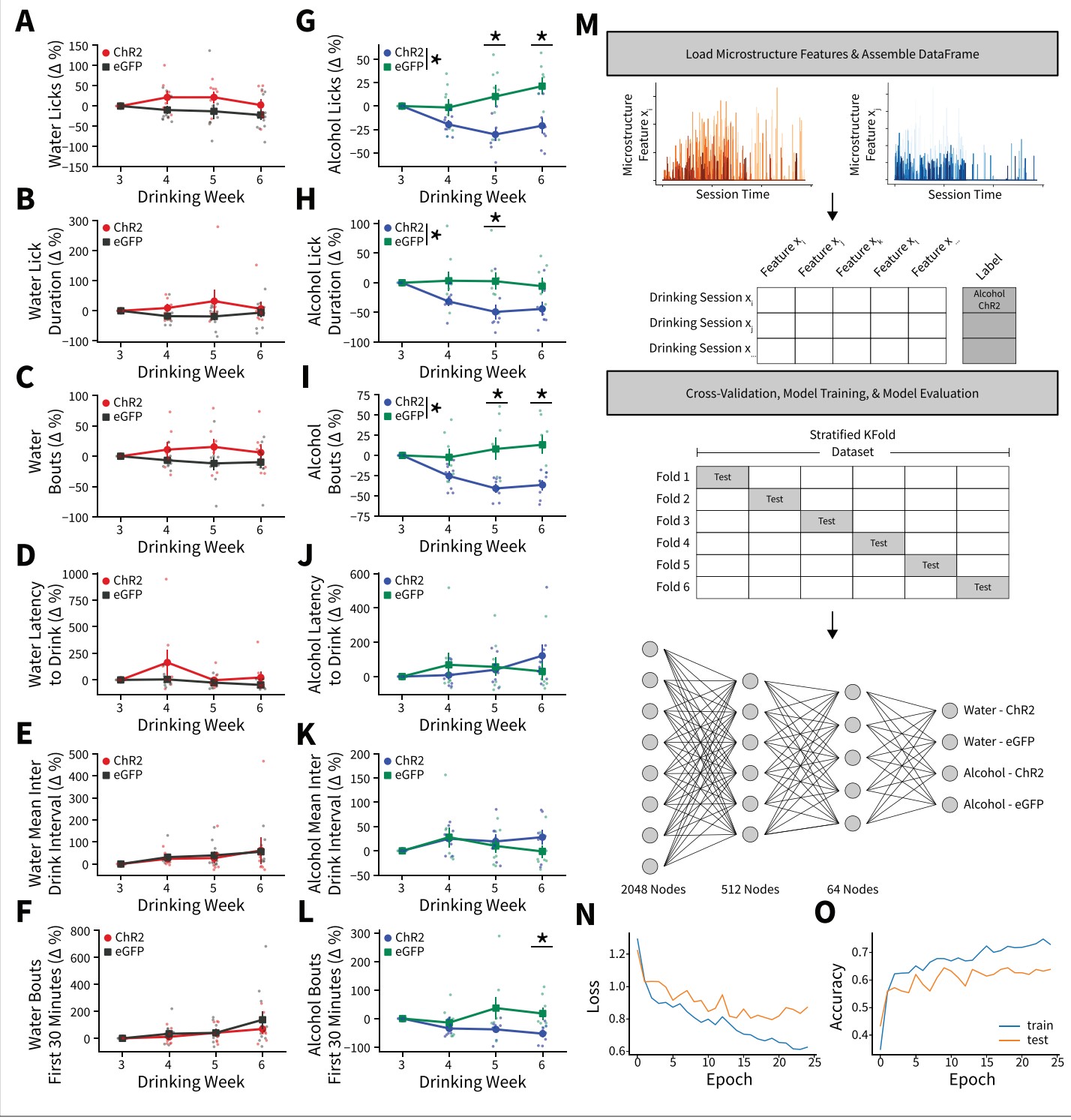

**Figure 3.** Alterations in drinking microstructure represent significant decreases in alcohol intake and are predictive of AIC→DLS alcohol-induced synaptic plasticity changes. Photoexciting anterior insular cortex (AIC) inputs does not alter the (**A**) number of water licks (two-way mixed analysis of variance [ANOVA], Virus $F_{(1,17)}$ = 3.7529, p=0.0695; ChR2: 32 observations, n=8 animals, eGFP: 44 observations, n=11 animals), (**B**) water lick durations (two-way mixed ANOVA, Virus $F_{(1,17)}$ = 2.2136, p=0.1551; ChR2: 32 observations, n=8 animals, eGFP: 44 observations, n=11 animals), (**C**) water bouts (two-way mixed ANOVA, Virus $F_{(1,17)}$ = 3.0848, p=0.0971; ChR2: 32 observations, n=8 animals, eGFP: 44 observations, n=11 animals), (**D**) latency to drink water (two-way mixed ANOVA, Virus $F_{(1,17)}$ = 2.7012, p=0.1186; ChR2: 32 observations, n=8 animals, eGFP: 44 observations, n=11 animals), (**E**) mean inter-drink-interval for water drinking (two-way mixed ANOVA, Virus $F_{(1,17)}$ = 0.0272, p=0.8708; ChR2: 32 observations, n=8 animals, eGFP: 44 observations, n=11 animals), (**F**) or the number water bouts in the first 30 min of the Drinking in the Dark (DID) session (two-way mixed ANOVA, Virus $F_{(1,17)}$ = 0.5716,

*Figure 3 continued on next page*

*Figure 3 continued*

p=0.4599; ChR2: 32 observations, n=8 animals, eGFP: 44 observations, n=11 animals). Photoexciting AIC inputs (**G**) decreases the number of alcohol licks (two-way mixed ANOVA, Virus $F_{(1,14)}$ = 11.0142, p=0.0051; Virus × Drinking Week $F_{(3,42)}$ = 5.8888, p=0.0019; Week 5 p=0.0307, Week 6 p=0.0137; ChR2: 32 observations, n=8 animals, eGFP: 32 observations, n=8 animals), (**H**) decreases total alcohol lick duration (two-way mixed ANOVA, Virus $F_{(1,14)}$ = 6.9536, p=0.0195; Virus × Drinking Week $F_{(3,42)}$ = 3.8533, p=0.0160; Week 5 p=0.0458; ChR2: 32 observations, n=8 animals, eGFP: 32 observations, n=8 animals), and (**I**) decreases the number of alcohol bouts (two-way mixed ANOVA, Virus $F_{(1,14)}$ = 11.2086, p=0.0048; Virus × Drinking Week $F_{(3,42)}$ = 9.4893, p=0.0001; Week 5 p=0.0135, Week 6 p=0.0051; ChR2: 32 observations, n=8 animals, eGFP: 32 observations, n=8 animals). (**J**) Modulating AIC inputs does not impact the latency to drink alcohol (two-way mixed ANOVA, Virus $F_{(1,14)}$ = 0.0084, p=0.9281; ChR2: 32 observations, n=8 animals, eGFP: 32 observations, n=8 animals) or (**K**) the mean inter-drink-interval for alcohol drinking (two-way mixed ANOVA, Virus $F_{(1,14)}$ = 0.4686, p=0.5047; ChR2: 32 observations, n=8 animals, eGFP: 32 observations, n=8 animals), but (**L**) decreases front loading behaviors for alcohol drinking (two-way mixed ANOVA, Virus $F_{(1,14)}$ = 4.2003, p=0.0596; Virus × Drinking Week $F_{(3,42)}$ = 3.9078, p=0.0150; Week 6 p=0.0230; ChR2: 32 observations, n=8 animals, eGFP: 32 observations, n=8 animals). (**M**) Schematic for microstructure feature detection, dataset assembly, cross-validation, and network architecture. (**N**) Loss curve visualization for training and testing data. (**O**) Model accuracy for training and testing data. Error bars indicate ± SEM. All post hoc comparisons are Sidak corrected.

The online version of this article includes the following source data for figure 3:

**Source code 1.** Alterations in drinking microstructure represent significant decreases in alcohol intake and are predictive of AIC→DLS alcohol-induced synaptic plasticity changes.

**Source data 1.** Alterations in drinking microstructure represent significant decreases in alcohol intake and are predictive of AIC→DLS alcohol-induced synaptic plasticity changes.

a cohort of alcohol-naive animals, we performed a light-dark test (*Figure 4D*). There were no differences between ChR2 and eGFP controls in time spent on the light side of the box across LED epochs, total number of light side entries, or latency to enter the dark side (*Figure 4E–G*). Finally, we used optogenetic intercranial self-stimulation to determine if modulating AIC inputs itself was reinforcing (*Figure 4H*). There were no group differences between nosepoking behaviors for photoexcitation (*Figure 4I and J*). Altogether, these data indicate that modulating AIC inputs within the DLS does not decrease alcohol intake via changes in reward perception, valence, reinforcement, locomotion, or anxiety-like behavior.

In conclusion, we have identified and characterized how binge consumption of alcohol induces changes in synaptic plasticity in a novel corticostriatal circuit that behaviorally governs binge alcohol drinking. Here, we used unilateral photoexcitation of AIC inputs to the DLS. Given that unilateral and bilateral striatal stimulation can have different outcomes on behavior, future work could explore whether the same holds true for bilaterally activating AIC→DLS synapses (*Kravitz et al., 2010*). Future work is also required to determine the mechanisms whereby alcohol-induced neuroplasticity alters circuit function to produce changes in drinking and what specific components of drinking behavior AIC→DLS synapses govern, especially how these mechanisms may differ by biological sex.

Nonetheless, the presented data suggest that there exist alcohol-induced synaptic changes within this circuitry that may serve as a critical biomarker for identifying binge alcohol consumption that leads to the future development of AUD. This identification not only will help advance basic substance use research, but may also provide translational value in the prevention and clinical treatment of AUD (*Lovinger and Gremel, 2021*). Specifically, the existence of a measurable circuit change that tracks with an increased number of binge drinking episodes could be used as the basis for a more reliable quantification of alcohol use and binge drinking history. The presence or absence of these circuit changes may also aid clinicians and researchers in determining a more accurate probability of when or if an individual is at an elevated risk of developing AUD. The usefulness of both measures helps create new knowledge that can foster preventative treatment approaches and helps further identify alcohol consumption patterns that may put an individual at increased risk of developing AUD (*Greenfield et al., 2014*). More accurately determining AUD risk can be a difficult task which has a biological basis, but is also stratified by social, societal, religious, and cultural factors that can make that calculation difficult. Providing more diverse biological measurements has proven beneficial in improving this risk calculation (*Baggio et al., 2020*).

New therapeutic modalities, such as non-invasive applications of brain stimulation, like transcranial magnetic stimulation (TMS) are already being investigated in humans to treat AUD. Current approaches for TMS aimed at treating AUD have provided mixed results (*Perini et al., 2020*; *McCalley et al., 2022*; *Tang et al., 2021*). The anatomical location of the anterior parts of the insular cortex in humans

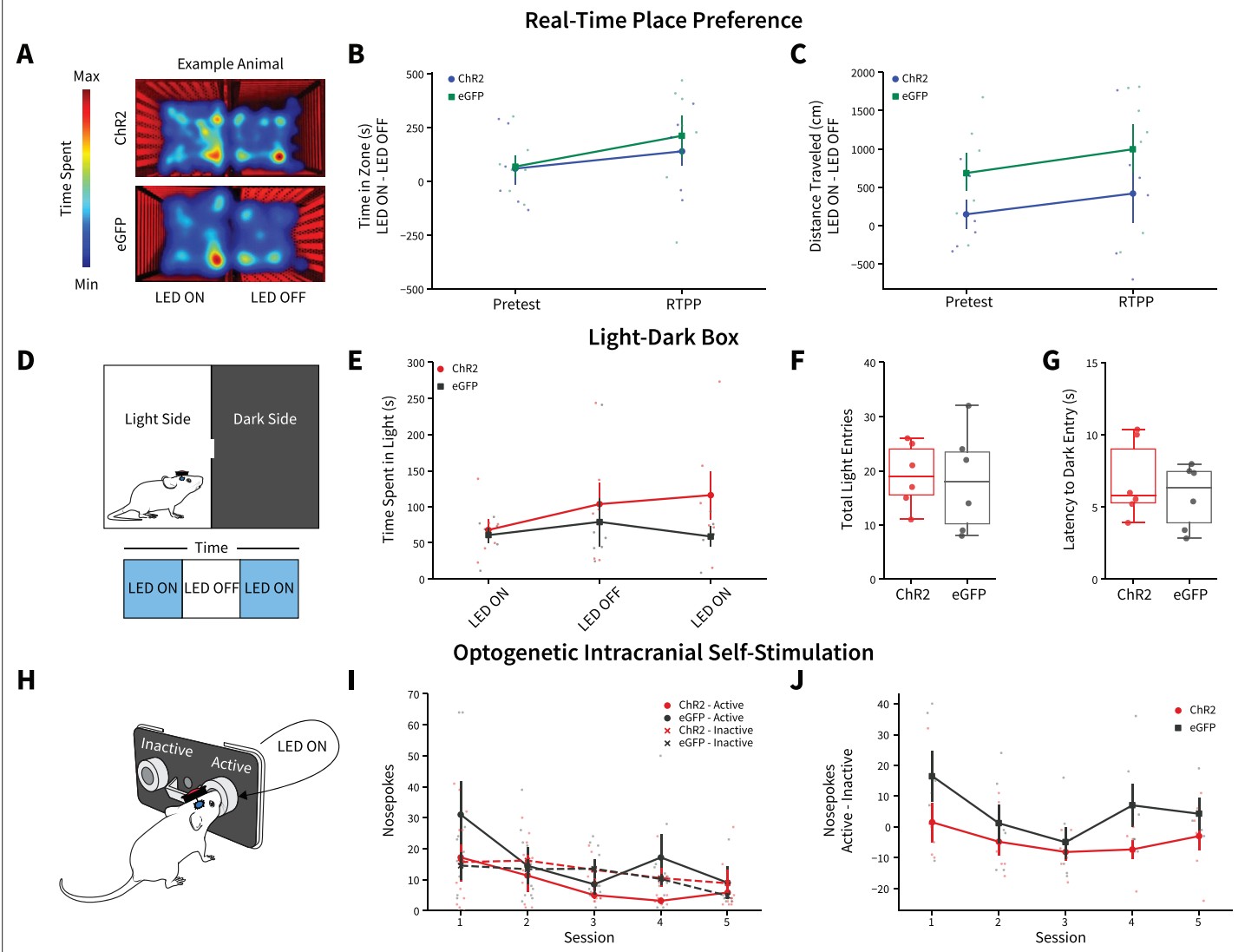

**Figure 4.** Anterior insular cortex (AIC) inputs to the dorsolateral striatum (DLS) do not modulate valence, anxiety-like, or operant responding behaviors. (**A**) Example animal heatmap for real-time place preference assay. Photoexciting AIC inputs within the DLS did not alter (**B**) real-time place preference/avoidance behaviors (two-way mixed analysis of variance [ANOVA], Virus $F_{(1,11)}$ = 0.2330, p=0.6388; ChR2: n=6 animals, eGFP: n=7 animals) or (**C**) locomotion during real-time place preference (two-way mixed ANOVA, Virus $F_{(1,11)}$ = 3.8457, p=0.0757; ChR2: n=6 animals, eGFP: n=7 animals). (**D**) Schematic for light-dark box assay. Modulating AIC inputs did not alter (**E**) time spent on light side across LED epochs two-way mixed ANOVA, Virus $F_{(1,9)}$ = 1.6826, p=0.2268; ChR2: n=6 animals, eGFP: n=6 animals, (**F**) the number of total light side entries (t test, $t_{(10)}$ = 0.2204, p=0.83; ChR2: n=6 animals, eGFP: n=6 animals), or (**G**) delays to enter the dark (t test, $t_{(10)}$ = 0.7699, p=0.4591; ChR2: n=6 animals, eGFP: n=6 animals). (**H**) Schematic of operant self-stimulation testing. (**I**) Photoexciting AIC inputs did not alter nosepoking behaviors across session (three-way ANOVA, Virus × Poke × Session $F_{(4,96)}$ = 0.3108, p=0.8702; ChR2: n=6 animals, eGFP: n=6 animals). (**J**) Active–inactive nosepokes do not differ between viral expression when modulating AIC (two-way mixed ANOVA, Virus $F_{(1,8)}$ = 5.0255, p=0.0553; study sufficiently powered to detect main effect of Virus as determined by power analysis, $\beta$=0.9898; ChR2: n=6 animals, eGFP: n=6 animals). Error bars indicate ± SEM. Box plot whiskers represent interquartile range.

The online version of this article includes the following source data for figure 4:

**Source data 1.** Anterior insular cortex (AIC) inputs to the dorsolateral striatum (DLS) do not modulate valence, anxiety-like, or operant responding behaviors.

are also easily accessible by non-invasive approaches and targeted therapies to this brain region may help advance new non-abstinence-based therapeutic approaches to aid those individuals' seeking treatment for AUD (*Ekhtiari et al., 2019*). The possibility that current and next-generation targeted therapeutics focused on circuit-based psychiatric approaches for AUD can modulate this specific brain

circuitry to decrease binge drinking behaviors in humans has the potential to immensely impact public health and increase the overall health and quality of life for millions that suffer from AUD.

# Materials and methods

## Animals

Animal care and experimental protocols for this study were approved by the Institutional Animal Care and Use Committee at the Indiana University School of Medicine (IACUC #: 19017) and all guidelines for ethical protocols and care of experimental animals established by the National Institutes of Health (Bethesda, MD) were followed. Male C57BL/6J mice were ordered from the Jackson Laboratory (Bar Harbor, ME). Animals arrived at 6 weeks of age and were allowed to acclimate to the animal facility for at least 1 week before any surgery was performed. All animals were group-housed (except for mice that underwent DID experiments as outlined below) in a standard 12 hr light/dark cycle (lights on at 08:00 hr). Humidity in the vivarium was held constant at 50% and food and water were available ad libitum.

## Stereotaxic surgeries

All surgeries were conducted under aseptic conditions using an ultra-precise small animal stereotaxic instrument (David Kopf Instruments, Tujunga, CA). Animals were anesthetized using isoflurane (3% for induction, 1.0–1.5% for maintenance at a flow rate of 25–35 ml/min of oxygen depending on body weight at the time of surgery). Viral injections were performed using a 33-gauge microinjection needle (Hamilton Company, Reno, NV). Animals were treated post-operatively for 72 hr with daily injections of carprofen (5 mg/kg) and topical lidocaine on the surgical incision. Animals were allowed to recover for at least 1 week before behavioral assays began. Animals were assigned to groups randomly and after surgery animal IDs were re-coded to blind experimenters to viral expression status.

For electrophysiology experiments, all mice were injected bilaterally with AAV9-*CaMKIIa*-ChR2(H134R)-EYFP (Addgene, 26969-AAV9) to drive ChR2 expression in the AIC at coordinates A/P: +2.4, M/L: ±2.30, D/V: −3.00 (50 nl/injection, 25 nl/min infusion rate).

For wireless in vivo optogenetic experiments, all mice were injected bilaterally with AAV9-*CaMKIIa*-ChR2(H134R)-EYFP (Addgene, 26969-AAV9) or AAV9-*CaMKIIa*-eGFP (Addgene, 105541-AAV9) to drive ChR2 expression or eGFP control in AIC at coordinates A/P: +2.4, M/L: ±2.30, D/V: −3.00 (50 nl/injection, 25 nl/min infusion rate). To modulate AIC inputs in the DLS, mice were unilaterally implanted with a wireless probe (4 mm depth, 470 nm blue light LED – Neurolux, Northfield, IL) at coordinates A/P: +0.7, M/L: ±1.55, D/V: −2.5. The LED orientation allowed for light to travel from anterior DLS to posterior DLS and ensured minimization of off-target light activation of AIC cell bodies while maximizing light coverage for AIC input innervation in the DLS. Probes were placed in either the right or left DLS, but there was no lateralization effect on alcohol or water intake, thus data are presented collapsed on probe location. Probes were secured to the skull using ethyl cyanoacrylate (LOCTITE 444, Henkel, Rocky Hill, CT) and the skin was closed over the top of the probe using Vetbond tissue adhesive (3M, Saint Paul, MN).

## Drinking in the Dark

The DID paradigm was based on the original DID procedure (*Thiele and Navarro, 2014*) with two modifications. First, there were four 2 hr DID sessions (Monday–Thursday) and one 4 hr DID session (Friday) and second, all DID sessions were performed out of a single bottle of water or alcohol (20% v/v in water) via lickometers (described below) inserted into the cage at the beginning of the DID session, which were removed at the end of the DID session. Mice had ab libitum access to their standard water bottles at all other times.

To summarize the performed procedures, after animals recovered from surgery, they were singly housed and allowed to acclimate in a reverse 12 hr dark/light cycle (lights off at 06:00 hr) for 1 week. Three hours into the dark cycle (09:00 hr) the standard water bottle was removed from the cage and the lickometer was inserted into the cage after the fluid bottle was weighed for 2 or 4 hr depending on the test day. Following the completion of 2- or 4-hr access, the lickometers were removed and the fluid bottles were weighed immediately after the session. Grams per kg (g/kg) of water and alcohol were computed from the difference in bottle weight and the density of water or 20% alcohol in water.

Mice were weighed post-DID session on Monday, Wednesday, and Friday (Tuesday and Thursday weights were equivalent to Monday and Wednesday, respectively). For Saturday and Sunday, the mice had no access to the lickometers as no procedures were performed. These 1-week repeating DID cycles are referred to as 'Drinking Weeks'.

Water and alcohol intakes for study inclusion were determined by summing the intake across the first 15 DID sessions. Animals that fell outside of the lower interquartile range (IQR) for their respective fluid type were excluded from the study for not consuming enough water or alcohol in the binge acquisition period. Only one alcohol animal was removed from the study for failure to meet this criterion as it only consumed a total of 1.28 g/kg of alcohol across all 15 DID sessions.

## Blood alcohol concentration

In a subset of animals following all DID sessions as to not disturb drinking, retro-orbital blood samples were taken after an additional 2-hr DID session. BACs were determined via gas chromatography (GC-2010 plus, Shimadzu, Japan) and correlated with respective alcohol intakes from that session.

## Electrophysiology

Brain slices were prepared 24 hr after the completion of the mice's final sessions during week 4 (sessions 16–20) of DID. Mice were anesthetized with isoflurane and then immediately killed via decapitation. The brain was rapidly dissected out and placed in an ice-cold cutting solution containing (in mM): 194 sucrose, 30 NaCl, 26 NaHCO$_3$, 10 glucose, 4.5 KCl, 1.2 NaH2PO$_4$, and 1 MgCl$_2$, saturated with a 95/5% O$_2$/CO$_2$ mixture. Brain slices containing the DLS were made at a thickness of 280 µm on a VT1200S vibratome (Leica, Germany). Slices were incubated in artificial cerebrospinal fluid (aCSF) containing (in mM): 124 NaCl, 26 NaHCO$_3$, 10 glucose, 4.5 KCl, 2 CaCl$_2$, 1.2 NaH$_2$PO$_4$, 1 MgCl$_2$ (310–320 mOsm) saturated with a 95/5% O$_2$/CO$_2$ mixture at 30°C for 1 hr after which they were moved to incubate at room temperature until recording. DLS brain slices were transferred to a recording chamber continuously perfused with 95/5% O$_2$/CO$_2$-saturated aCSF solution at a rate of 1–2 ml/min at 29–32°C.

DLS brain slices were visualized on a BX51WI microscope (Olympus Corporation of America, Center Valley, PA). Whole-cell patch clamp electrophysiological recordings were made from MSNs in the voltage clamp configuration. Recordings were made using a Multiclamp 700B amplifier and a 1550B Digidata digitizer (Molecular Devices, San Jose, CA). Glass patch pipettes (borosilicate with filament, 2.0–3.5 MΩ resistance, World Precision Instruments, Sarasota, FL) were made using a P-1000 micropipette puller (Sutter Instruments, Novato, CA). Pipette internal solution contained (in mM) 120 CsMeSO$_3$, 10 HEPES, 10 TEA-Cl, 5 lidocaine bromide, 5 NaCl, 4 Mg-ATP, 1.1 EGTA, and 0.3 Na- GTP (pH 7.2, 290–310 mOsm). MSN identity was determined visually based on soma size and confirmed via measures of membrane resistance and capacitance. MSNs were constantly voltage clamped at −60 mV (except when recording NMDA receptor-mediated currents), using sampling rate of 10 kHz and low-Bessel filter 2 kHz. No correction for liquid junction potential was used. In order to isolate excitatory transmission, picrotoxin (50 µM) was added to the aCSF solution for whole-cell recordings and a subset of field recordings. Extracellular field recordings were used to measure network responses using glass micropipettes filled with 1 M NaCl. Both whole-cell patch pipettes and field recording pipettes were placed in the regions of the DLS where AIC innervation was the most dense (*Figure 1—figure supplement 1*) for recordings. Data were acquired using Clampex 10.3 (Molecular Devices, San Jose, CA). For whole-cell recordings, series resistance was monitored throughout the experiment. Cells with series resistances greater than 25 MΩ or that changed more than 15% during recording were excluded from data analyses.

Optically evoked excitatory postsynaptic currents (oEPSCs) or oPSs were produced in DLS brain slices using 5 ms 470 nm blue LED light pulses delivered via field illumination through the microscope objective. oEPSCs and oPSs were evoked every 30 s. Stimulus-response measures of oEPSC were performed by increasing the intensity of blue LED light from 0 up to 1 mW. For AMPA/NMDA and paired-pulse ratio (PPR) whole-cell recording measures, light intensity was initially adjusted to produce stable oEPSCs of 100–600 pA amplitude after which experimental recordings were begun. oAMPA/oNMDA current ratio recordings were measured by first holding the cell at −80 mV and optically evoking an AMPAR-mediated EPSC. The NMDAR current was then determined by shifting the membrane potential to +40 mV and optically evoking an EPSC. Since the AMPAR component of

oEPSCs at −80 mV was not apparent at 100 ms following the optical stimulus (the measured current rapidly returned to baseline), we calculated the NMDAR-mediated portion of the oEPSC at +40 mV as the average of the measured current over the following 25 ms (100–125 ms post-stimulus), similar to our previously published protocol (*Fritz et al., 2018*). oPPRs were measured by performing two optical stimulations separated by 50–200 ms averaging three technical replicates. oPPR was calculated by taking the average amplitude of the second pulses (using tail current as baseline) and dividing it by the average amplitude of the first pulses (using pre-stimulation current as baseline). Non-optically evoked sEPSCs were also measured using whole-cell voltage clamp recordings, with MSNs held at −60 mV using sampling rate of 50 kHz. sEPSCs were measured over the course of a 2 min gap-free recording. sEPSC amplitude, frequency, and decay and rise times were computed using Minianalysis (Synaptosoft, Decatur, GA). Stimulus-response measures of oPSs were performed by increasing the intensity of blue LED light from 0 up to 1.15 mW. To test whether the optical stimulation used in vivo produced plasticity on its own, we recorded a 10 min baseline of oPS amplitudes elicited every 30 s after which we optically stimulated the brain slices with blue LED light (again, through the objective) at 20 Hz for 10 s. oPS amplitudes were then measured again at 30 s intervals. In one set of experiments, at the conclusion of recording the effects of the 20 Hz stimulation, the AMPA receptor antagonist NBQX (5 µM) was bath applied to the brain slice. In a separate set of experiments, the same recordings of oPS amplitudes with 20 Hz stimulation were performed but in the presence of the GABAA receptor antagonist picrotoxin (50 µM).

## Histology

Animals were anesthetized with isoflurane and trans-cardially perfused with 15 ml of ice-cold phosphate buffered saline (PBS) followed by 25 ml of ice-cold 4% paraformaldehyde (PFA) in PBS. Animals were decapitated and the brain was extracted and placed in 4% PFA for 24 hr. Brains were then transferred to 30% sucrose solution until they sank after which they were sectioned on a vibratome at a section thickness of 50 µm. Brain sections were mounted in serial order on glass microscope slides and stained with 4',6-diamidino-2-phenylindole to visualize nuclei. Fluorescent images were captured on a BZ-X810 fluorescent microscope (Keyence, Itasca, IL) using ×4 and ×20 air objectives. Injection site, viral expression, and the location of optogenetic probe implantations were determined from matching images to the Reference Allen Mouse Brain Atlas. Animals that did not have viral expression in the brain regions of interest and/or did not have optogenetic probe placements within the brain region of interest as confirmed by histology were excluded from the study.

## Lickometers

All alcohol and water drinking experiments were performed out of custom-built homecage fluid monitors (i.e. 'lickometers') that were constructed as described (*Godynyuk et al., 2019*) with the following modifications. Liquid monitoring was constantly sampled as a function of the state (open vs. closed) of an infrared beam directly in front of the fluid bottle valve, but data was only written to device memory at minimum every 2 s. Therefore, any tube interaction within a 2 s window (defined as a drinking bout) was recorded as the total number of beam breaks (defined as licks) for the total duration that the beam was broken (defined as lick duration) within that bout. Drinking bouts could be longer than 2 s, but never shorter.

The lickometers can hold two bottles, although we only used one for each device during these experiments. We randomized the bottle side location across animals and found no effect on drinking, thus data are presented collapsed on bottle side.

Lickometer data was cleaned by fitting a linear model comparing the number of licks by the lick duration within each bout for every DID session performed by each animal.

Bouts that had a residual value of greater than or less than 3 from model fit were removed. This cleaning procedure removed bouts that were due to slow leaks (a new tube was used if a leak was detected for the next DID session) or chews on the bottles. A total of 3,928,956 bouts were recorded across all experiments, and after cleaning 3,927,570 remained (99.9647% of bouts remained) showing that almost all bottle contacts were directed drinking behaviors, and that chews and leaks were rare.

For wireless in vivo optogenetic experiments that utilized lickometers for closed-loop control during DID sessions, the lickometer's BNC out was connected to a TTL port on the Neurolux optogenetics system such that when a lick occurred, for the entire duration of the lick, a voltage change was

sent from the lickometer to turn on the optogenetics probe. Optogenetic stimulation was delivered at 20 Hz with 5 mW of light power at 470 nm for the entire duration of the beam break.

In a subset of animals, we also performed yoked controls such that yoked animals received optogenetic stimulation at stochastic time points during DID sessions. The yoked controls' stimulations were dependent on the drinking activity of some other animal in the room during the DID session to ensure the amount of stimulation and type (length, interval, etc.) was similar across groupings. There were no differences in intake by yoked status for water or alcohol, thus the data are presented collapsed on yoked and closed-loop stimulation. This finding suggests that changes in alcohol intake driven by optogenetic manipulations may be generalizable to when alcohol is accessible or at timepoints near alcohol consumption behaviors, and not necessarily linked to closed-looped stimulation during licking activity.

## Microstructure feature analyses

Using licks, lick duration, bouts, and the timestamps when these events occurred during the DID session, we calculated other drinking features such as latency to drink (time to first bream break after session initiation) and mean inter-drink interval time per session (the mean time between bouts within each DID session). We also calculated features for events in the first 30 min of each DID session as a measure of front-loading behavior. Finally, we used maximum values (max bout length, max licks per bout, etc.) to characterize microstructure features within DID sessions.

## Machine learning

A feedforward artificial neural network was used to determine if microstructure features from a single DID session could predict the experimental manipulations across fluid type (alcohol vs. water) and viral expression (ChR2 vs. eGFP). To bolster our predictive capabilities, we used all 18 microstructure features we computed per DID session.

Features were concatenated across all animals and DID sessions into a single data frame and categorical labels were one-hot encoded. Data were normalized to a range of 0–1 before cross-validation using stratified k-fold (k=6) to ensure the unequal proportion of fluid × virus labels in the dataset did not influence model training predictions. Each shuffled fold contained 832 training examples and 166 testing samples. The sequential network architecture contained 2048 nodes in the first layer, 512 nodes in the second layer, 64 nodes in third layer, and an output layer of 4 nodes. We used a rectified linear unit activation function for the first three layers and a softmax activation function for the final output layer. The model was compiled using an Adam optimizer and loss was scored using categorical cross entropy. After training all six folds, the average testing accuracy on data previously unseen by the model was 66.13% with a maximum of 70.48% accuracy for one of the training folds.

## Real-time place preference

One week after the final DID session (session 30), animals were subjected to an RTPP assay to determine if photoexciting AIC inputs altered valence states that could explain differences in alcohol behaviors. Animals were placed in an arena (40 cm × 20 cm × 30 cm) with a divider down the middle that contained a cutout so mice could freely pass through from one side to the other. One side of the arena contained a mesh floor with vertical striped walls and the other with wire floor and horizontal striped walls. All testing was performed in the dark using red light and infrared cameras to capture behavioral videos. Two boxes were used, each with counterbalanced LED ON sides that triggered the activation of the optogenetic probe to deliver 20 Hz 470 nm light pulses at 5 mW when entry into that side of the arena occurred for the entire duration the animal was in that side of the arena. For the pretest day, animals were placed in the arena for 20 min with no optogenetic activation regardless of their location in the arena. On the RTPP testing day the same procedure was repeated except the LED ON side of the box triggered optogenetic stimulation. All location and locomotion data were computed using EthoVision XT (V15 – Noldus, Skokie, IL) from a camera placed directly above the arena which measured the center of the animal as it freely behaved across testing.

## Light dark box

A light dark box assay was used in a separate cohort of alcohol-naïve mice to determine if photoexciting AIC inputs could alter anxiety-related behaviors. Mice were placed into an arena (40 cm × 40 cm

× 30 cm) that was split in half with a covered divider that contained a cutout so mice could freely pass through from one side to the other. The open side of the box was brightly lit (300–500 lumens) with overhead light and the other covered side registered dimly lit (0–10 lumens). The 15 min session consisted of three epochs (LED ON, LED OFF, LED ON) for 5 min each. During the LED ON epochs, optogenetic stimulation was delivered at 20 Hz with 470 nm light pulses at 5 mW for the duration of the epoch. Animals were placed in the corner furthest from the dark entry to begin the assay. All animal tracking data was computed using EthoVision XT (V15 – Noldus, Skokie, IL) from a camera placed directly above the arena which measured the center of the animal as it freely behaved across testing.

### Intracranial self-stimulation

Intracranial self-stimulation was used to determine if photoactivation of AIC inputs could maintain an operant response. Intracranial self-stimulation was performed in a homecage (30 cm × 15 cm × 15 cm) with a FED3 device (*Matikainen-Ankney et al., 2021*). Each FED3 device has two nose poke ports ('active' and 'inactive'). Responding in the active port was reinforced with photoactivation on a fixed-ratio 1 schedule and resulted in light delivery (5 mW of 470 nm light in 25 ms pulses for a total of 20 pulses) as well as presentation of a tone (4 kHz for 300 ms) and illumination of a cue light bar located below the active nose port. During only the initial session, the back of the active nose poke was baited with crushed Froot Loops and sucrose pellets to encourage nose poking behaviors. The inactive port resulted in no photoactivation, tone, or light cue. Animals were run for five sessions, each 1 hr long and the active port was randomized and balanced across all animals.

### Modeling and statistics

Sample sizes for all experiments were determined based on previously published experimental findings for electrophysiology and in vivo behavioral assays such as DID (*Muñoz et al., 2018*).

Data preprocessing and machine learning modeling utilized SciPy (*Virtanen et al., 2020*), Statsmodels (*Seabold and Perktold, 2010*), Scikit-learn (*Pedregosa et al., 2011*), and TensorFlow (*Abadi et al., 2016*). Statistical analyses were performed using GraphPad Prism (GraphPad Software – V9.2, San Diego, CA) and pingouin (V0.5.0, *Vallat, 2018*). Data visualization used matplotlib (*Hunter, 2007*) and seaborn (*Waskom, 2021*) libraries. For time series or repeated measures, we used two-way mixed ANOVAs, which represented time (session or week) or the repeated factor (light power, etc.) as the repeated, within-subject variable, and the factor (fluid type, viral expression, bottle side, etc.) as the between-subjects factor. All two-way mixed ANOVA data were tested to see if variances between factors were equal and normal using the Levene test. If there was a main effect for factor, we used pairwise t-tests to determine post hoc significance and p-values were Sidak corrected. For correlations between two variables, we tested multivariate normality using the Henze-Zirkler test. If samples were normal, we used a Pearson's correlation to report r and p-values. If samples failed normality testing, we used a Shepherd pi correlation that returned the Spearman correlation after removing bi-variate outliers. For tests of two factors, we used unpaired Welch two-tailed t-tests to correct for unequal sample sizes or paired t-tests. All significance thresholds were placed at $p<0.05$ and all data and model fits are shown as mean ± standard error (68% confidence interval) of the mean (SEM) or by box plot with error bars indicating the IQR. Data points beyond IQRs are represented as diamonds.

### Data and materials availability

All experimental data are available in the main text or the supplementary materials. Microstructure analysis and machine learning code is available at https://github.com/dlhagger/AIC-DLS_Microstructure_Modeling (*Haggerty, 2022*; copy archived at swh:1:rev:ccf174d370b21d632426e929f9bcfd5b539b211c).

## Acknowledgements

We thank Kaitlin C Reeves, Brandon M Fritz, and Fuqin Yin for experimental assistance and Erin A Newell for technical assistance. We also thank Lex Kravitz and Ethan Tyler for modified drawings that were accessed via scidraw.io. National Institutes of Health grant R01AA027214 (BKA). National Institutes of Health grant F31AA029297 (DLH). National Institutes of Health grant F30AA028687 (GGG).

National Institutes of Health grant T32AA07462 (DLH). Stark Neurosciences Research Institute (BKA, DLH, GGG). Indiana University Health (BKA).

## Additional information

### Funding

| Funder | Grant reference number | Author |
|---|---|---|
| National Institutes of Health | R01AA027214 | Brady K Atwood |
| National Institutes of Health | F31AA029297 | David L Haggerty |
| National Institutes of Health | F30AA028687 | Gregory G Grecco |
| National Institutes of Health | T32AA07462 | David L Haggerty |
| Indiana University Health | | Brady K Atwood |
| Indiana University-Purdue University Indianapolis | Stark Neurosciences Research Institute | David L Haggerty Gregory G Grecco Brady K Atwood |

The funders had no role in study design, data collection and interpretation, or the decision to submit the work for publication.

### Author contributions

David L Haggerty, Conceptualization, Resources, Data curation, Software, Formal analysis, Funding acquisition, Investigation, Visualization, Methodology, Writing - original draft, Project administration, Writing – review and editing; Braulio Munoz, Conceptualization, Data curation, Formal analysis, Methodology, Writing – review and editing; Taylor Pennington, Data curation, Formal analysis, Methodology; Gonzalo Viana Di Prisco, Data curation; Gregory G Grecco, Conceptualization, Software, Formal analysis, Funding acquisition, Validation, Investigation, Methodology, Writing – review and editing; Brady K Atwood, Conceptualization, Supervision, Funding acquisition, Methodology, Project administration, Writing – review and editing

### Author ORCIDs

David L Haggerty ⓘ http://orcid.org/0000-0002-1455-2557
Gregory G Grecco ⓘ http://orcid.org/0000-0002-0700-8633
Brady K Atwood ⓘ http://orcid.org/0000-0002-7441-2724

### Ethics

Animal care and experimental protocols for this study were approved by the Institutional Animal Care and Use Committee at the Indiana University School of Medicine (IACUC #: 19017) and all guidelines for ethical protocols and care of experimental animals established by the National Institutes of Health (Maryland, USA) were followed.

### Decision letter and Author response

Decision letter https://doi.org/10.7554/eLife.77411.sa1
Author response https://doi.org/10.7554/eLife.77411.sa2

## Additional files

### Supplementary files

• Transparent reporting form

## Data availability

Source data files are provided for all electrophysiology and behavior studies and machine learning model code is provided to reproduce all neural network analyses. Further code to replicate figures is hosted on GitHub at https://github.com/dlhagger/AIC-DLS_Microstructure_Modeling (copy archived at swh:1:rev:ccf174d370b21d632426e929f9bcfd5b539b211c).

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
