## [Editor Report]

Haggerty et al. reported findings examining how changes in brain function are involved in alcohol binge drinking, with a selective focus on the synaptic and circuit alterations that occur in the anterior insular cortex (AIC) inputs within the dorsolateral striatum (DLS). They show that chronic alcohol drinking produces glutamatergic synaptic adaptations in male mice and by stimulating this circuit binge drinking could be reduced without altering either water consumption or general performance for select reinforcing, anxiogenic or locomotor behaviors. The results of this study may specifically improve our understanding of the sex-specific differences in neurocircuitry mediating excessive drinking associated with alcohol use disorder.

---

## [Decision Letter]

**Decision letter after peer review:**

Thank you for submitting your article "Anterior insular cortex inputs to the dorsolateral striatum govern the maintenance of binge alcohol drinking" for consideration by *eLife*. Your article has been reviewed by 3 peer reviewers, and the evaluation has been overseen by a Reviewing Editor and Kate Wassum as the Senior Editor. The following individual involved in the review of your submission has agreed to reveal their identity: Andrew Lawrence (Reviewer 2).

Essential revisions:

We ask that you please address all reviewer comments and especially attend to the follow.

1. In figure 4J where n=2, please increase the n to at least 6/7 or provide power analysis justifying the n used if less than 6. See Reviewer 3 comment 3 and Reviewer 1 comment 1 for further explanation.

2. To strengthen the case for the specificity of the effect in figure 1 please repeat the experiments in Figure 1 to include female mice.

3. Please restrict the implication of the findings to addiction or the impact of these results to public health to the discussion only. In the discussion further expand on the use of AIC to DLS alterations as a biomarker for the development of alcohol use disorder.

4. For figure 1B, please clarify in the results and methods section whether brain slices were taken 24 or 48 hours after the final alcohol session. See Reviewer 2 comment 2 for further explanation.

4b. Also please report the rationale for starting the ex-vivo recording 24 h after the last binge episode. See Reviewer 1 comment 2 for further explanation.

5. Please add DAPI to the representative picture of viral injection in figure 1A. See Reviewer 1 comment 6 for further explanation.

6. Please add statistical p-values for figure 4C. See Reviewer 3 comment 3 for further explanation. Please also ensure your manuscript complies with the *eLife* policies for statistical reporting: https://reviewer.elifesciences.org/author-guide/full "Report exact p-values wherever possible alongside the summary statistics and 95% confidence intervals. These should be reported for all key questions and not only when the p-value is less than 0.05."

7. Please comment in the methods whether reported values include a correction for the liquid junction potential. See other comments section for reviewer 3.

8. Please clarify the description of the optogenetic stimulation paradigm in line 213.

See other comments section for reviewer 5.

*Reviewer 1 (Recommendations for the authors):*

Based on the comments above, these are the recommendations for the authors:

1. I strongly suggest increasing the number of mice to have at least 6/7 mice/group for each experiment.

2. Please report the rationale for starting the ex-vivo recording 24 h after the last binge episode. An alternative interpretation for the reported data could be that the glutamatergic synaptic adaptation observed are simply related to only one (i.e., the last) binge episode and not with the drinking history. Please discuss the possibility that these adaptations can occur with a single binge/alcohol exposure.

3. In line with the point above, it is not clear why for the electrophysiology recording the control virus is missing. This could be critical also for validating the behavioral approaches.

4. Regarding the issue with the unilateral stimulation, ideally the authors should repeat the experiments (at least some of the behavioral controls) to demonstrate similar results. Alternatively, the authors could add a "methodological consideration" section to explain these potential limitations.

5. I strongly suggest adding female mice to this manuscript to increase the impact of the results.

6. Please add DAPI to the representative picture of viral injection. It is difficult to appreciate anatomical landmarks for these pictures.

7. Please keep both introduction and discussion related to the circuit investigation and remove implications for human transition to addiction or the impact of these results on public health.

*Reviewer 2 (Recommendations for the authors):*

I enjoyed reading this manuscript and have a number of comments for the authors to consider.

1. In the introduction, please qualify the criteria of a binge as per NIAAA guidelines, namely 4 drinks in 2 hours for women and 5 drinks in 2 hours for men. Also, there is a recent review on the role of the insula in AUD that updates the cited Barker et al. 2015 review (see Campbell and Lawrence 2021).

2. In the text you state that brain slices are taken 24 hours after the final alcohol session; however, figure 1B implies that slices are taken 48 hours after alcohol. Which is it?

3. In rodents silencing insula inputs to the accumbens core reduces alcohol self-administration (Jaramillo et al., 2018), and in human heavy drinkers there is increased coupling between the right anterior insula and right nucleus accumbens (Grodin et al., 2018) during high threat alcohol cues. With regards the latter point, in rats the AI mediates relapse-like behavior in a punishment associated context (Campbell et al., 2019). Have the authors performed analogous recordings at AI→accumbens synapses to compare vs the AI→DLS synapses? Also, how much spread of virus was there along the A-P axis of the AI? In this regard rodent studies suggest that functionally inhibiting the caudal portion of the insular cortex reduced alcohol consumption whereas functionally inhibiting the anterior insular cortex. was without effect (Haaranen et al., 2020; Pushparaj and Le Foll, 2015).

4. The authors conclude that "binge drinking alcohol specifically potentiates postsynaptic AMPA and NMDA glutamate receptor responses at AIC→DLS synapses". There is evidence that AI→ventral striatum projections are involved in compulsive food eating (Spierling et al., 2020). As the prior comment, have the authors assessed AI→ventral striatum synapses? Also, while I acknowledge the authors previously showed the disruption of opioid receptor mediated LTD at AIC→DLS synapses was specific to alcohol, and did not occur after sucrose binge (Munoz et al., 2018), have they assessed the specificity of the current data set in relation to alcohol vs other (non-drug or drug) rewards?

5. Note for discussion, within the DLS in both humans and rodent models, muscarinic M4 receptors are subjected to alcohol-induced adaptations and implicated in alcohol seeking (Walker et al., 2020). Have the authors considered this possibility in terms of binge behavior?

6. Figure 2 G,H – why are the data presented as %? Moreover, in figure 2H does alcohol intake increase across weeks in the eGFP group?

7. Supplemental figure 7B – in the ChR2 group the baseline alcohol intake drops both times prior to the final two binge tests (see sessions 21-30). Does this impact the % change data? Over the same period the baseline intake in the eGFP group is more stable.

8. Supplemental figure 9 – the drinking profile of the LED off group is unstable, with intake seeming to escalate with time. Is this the case and if so why?

9. In terms of a molecular mechanism, have the authors considered investigating the potential role of BDNF / p75 (see Darcq et al., 2016)?

10. Binge drinking in adolescent females is a growing problem, have the authors studied female mice?

*Reviewer 3 (Recommendations for the authors):*

1) PPR at 25 ms should not be included. The very low values reflect poor fidelity of ChR2 at stimulation frequencies >30 Hz, rather than biological properties of AIC->DLS synapses. Also, the interpretation of these data in line 120 should use specific language to indicate directionality in place of "alter".

2) I suggest switching orders of graphs throughout so that the control groups (Water, eGFP, etc) precede the experimental groups (Alcohol, ChR2, etc)

3) I do not understand how the DID intake values for the Water group are tenfold higher than the Alcohol group. Also, "g/kg" is an unusual way to display Water Intake, I think "mL" is more informative and conventional.

4) The representative AMPA/NMDA ratio trace in Figure 1F displays significant AMPA receptor rectification. MSNs do not typically display significant AMPAR rectification under control conditions. Furthermore, the internal solution does not appear to contain spermine, so voltage dependent block of CP-AMPARs should not be a major factor. Did the authors correct for the liquid junction potential?

5) The description of the optogenetic stimulation paradigm (line 213) is misleading. As written, it reads as if most of the animals underwent closed-loop stimulation but only a minor portion were open-loop and eventually folded into the main data set. By contrast, the cohort was split n=4/4 open/closed as shown in Figure S6. This section should be re-written accordingly.

6) The authors should be commended for assessing the effects of their optogenetic stimulation on long-term plasticity in brain slices. Experiments like this are rarely performed but can provide useful, if not necessary, information for interpreting the effects of in vivo optogenetic stimulation. While no long-term effects were detected, it appears to me as if there could be an Alcohol x Picrotoxin interaction immediately after the 20 Hz stimulation. It seems as if picrotoxin might have decreased the post-tetanus potentiation in slices from Water but not Alcohol mice. If so, this could provide some insight into changes in AIC->DLS feedforward inhibition following DID. Also, these experiments should be described in greater detail in the Methods.

7) While the summarized oPS timecourses are completely blocked by NBQX, the representative oPS traces do not appear to be.

8) There seems to be an effect of AIC->DLS stimulation during the first DID cycle in Figure S9.

[Editors' note: further revisions were suggested prior to acceptance, as described below.]

Thank you for resubmitting your work entitled "Anterior insular cortex inputs to the dorsolateral striatum govern the maintenance of binge alcohol drinking" for further consideration by *eLife*. Your revised article has been evaluated by Kate Wassum (Senior Editor) and a Reviewing Editor.

The manuscript has been improved but there are some remaining issues that need to be addressed, as outlined below:

1. Amend the title of the manuscript to "The role of anterior insular cortex in binge alcohol drinking" (or similar).

2. Add to the main text that unilateral opto manipulation was used.

3. Include missing citation on Line 1002.

4. Add the "n" values to the figure 3 legend. Please double-check that all figure legends contain such "n" value information.

5. Perform power analysis to determine whether the ICSS dataset is adequately powered. State the result (whether or not ) concisely in the text. For example ( p-value, study sufficiently powered as determined by power analysis).

6. Add missing citations to the conclusion.

*Reviewer 3 (Recommendations for the authors):*

The addition of female mice is very welcome, and I agree that the lack of effects on physiology provides justification for limiting the behavioral studies to male mice. The introduction is much more thorough and provides greater context for the studies in this manuscript. The expanded conclusion is an improvement as well. I appreciate the additional replicates included in Figure 4 as well; however, I am not convinced that the ICSS experiment is fully powered.

The ICSS dataset, to me, appears to indicate a strong trend (.055) for AIC->DLS stimulation in decreasing discrimination of active vs inactive lever. This could be a very interesting finding in that it would suggest that the AIC->DLS pathway is aversive or regulates reward learning. Did the authors perform a power analysis for this experiment to be confident that N=6-7 is sufficient?

---

## [Author Response]

Essential revisions:We ask that you please address all reviewer comments and especially attend to the follow.1. In figure 4J where n=2, please increase the n to at least 6/7 or provide power analysis justifying the n used if less than 6. See Reviewer 3 comment 3 and Reviewer 1 comment 1 for further explanation.

Thank you for pointing out this weakness in our initial submission. For this revised manuscript we reran this experiment with 6 animals per group and updated Figure 4 I and J and the accompanying methods section titled “Intracranial self-stimulation” to reflect the change. We also note that the new, correctly powered experiment confirmed the previous claim that AIC inputs to the DLS do not modulate operant responding behaviors.

2. To strengthen the case for the specificity of the effect in figure 1 please repeat the experiments in Figure 1 to include female mice.

Thank you for this request. We took age-matched female mice that were injected with AAV-ChR2 into AIC and had them undergo the same 3 weeks of Drinking in the Dark to replicate the male data displayed in Figure 1 with an experimental focus on AIC inputs. We then performed whole cell patch clamp electrophysiology in DLS brain slices from these female mice. We measured optically evoked input-output responses (oEPSCs), AMPA/NMDA current ratios (oNMDA/oAMPA), and paired pulse ratios (oPPR). These data are presented in Figure 1—figure supplement 4. In contrast to males, we did not observe any effect of alcohol consumption on AIC inputs into the DLS of female mice compared to males. We also combined both male and female datasets to statistically determine if we had sex differences for these specific measures by the existence of a main effect and/or a sex x fluid interaction. We report these statistics in text from lines 189 to 204, where we note that we did not have a sex x fluid effect for oEPSCs but did note that we had a sex x fluid effect for our oNMDA/oAMPA synaptic plasticity measure. This finding further justifies the behavioral data and circuit manipulations being conducted in solely male mice.

While this is a fascinating sex difference and important data for the field, this manuscript is not specifically about exploring sex differences per se. We believe we have done our due diligence and correctly reported the existence of sex differences, or the possibility of sex differences, but the electrophysiological findings that we later modulate in vivo are only present in males. We point out that future work is needed to determine the contribution of circuit-specific changes in females at these synapses. Ultimately it will take much more work to fully elucidate sex difference circuit-specific mechanisms that we feel are far beyond the scope of this manuscript.

3. Please restrict the implication of the findings to addiction or the impact of these results to public health to the discussion only. In the discussion further expand on the use of AIC to DLS alterations as a biomarker for the development of alcohol use disorder.

We have removed language in the body of the manuscript and expanded on the implications of our findings at the end of our results and discussion from lines 553 to 587.

4. For figure 1B, please clarify in the results and methods section whether brain slices were taken 24 or 48 hours after the final alcohol session. See Reviewer 2 comment 2 for further explanation.4b. Also please report the rationale for starting the ex-vivo recording 24 h after the last binge episode. See Reviewer 1 comment 2 for further explanation.

We have revised the text to make it clear that this was 24 hours after the last exposure in lines 102104.

We decided the best way to address this comment was to perform an additional set of experiments where we took mice that had only one single alcohol drinking session and then recordings were again made 24 hours after that single exposure. We report these new findings in Figure 1—figure supplement 5.

We found that while a single session of alcohol DID exposure does produce a directionally similar change in oEPSCs amplitudes, changes in synaptic plasticity are not equivalent to those that occur following three weeks of DID. One single DID session produced an enhanced oEPSC response, the magnitude of which was larger than that produced by 3 weeks of DID, suggesting that while there are plastic changes of the same measure, their magnitudes (and therefore mechanisms) are longitudinally different. Furthermore, measurements of oNMDA/oAMPA ratios were not different across fluid exposure, nor were oPS amplitudes, suggesting the key measurements that we saw change after three weeks of DID were not sufficiently changed by a single DID exposure. Altogether these data argue that our prior findings are not due to simply a 24-hour withdrawal from alcohol, regardless of alcohol drinking history nor are they recapitulated by a single DID exposure. These data also support the differences in behavioral responses to AIC→DLS in vivo stimulation in mice with 3 weeks of alcohol drinking history compared to those that began receiving stimulation from the very first drinking session (Figure 2—figure supplement 6) which further speaks to the nature of repeated DID sessions having a longitudinal effect on plasticity at AIC inputs in the DLS.

5. Please add DAPI to the representative picture of viral injection in figure 1A. See Reviewer 1 comment 6 for further explanation.

We tried to post-fix an AIC injection from a brain that was used for an electrophysical experiments in DAPI, but due to limitations in slice thickness (~250 um) and the microscopes we have access to, the images were, in our opinion, worse than the example images provided. Since we have performed the same injections for electrophysiological experiments and in vivo experiments (in which we used a different method of histology to slice thinner sections, check hits, and stain with DAPI for which we provide an example image) and provided hit maps for all injections that show where in the brain the AIC is across bregma levels, there should not be an issue or confusion of what circuitry we are isolating.

6. Please add statistical p-values for figure 4C. See Reviewer 3 comment 3 for further explanation. Please also ensure your manuscript complies with the eLife policies for statistical reporting: https://reviewer.elifesciences.org/author-guide/full "Report exact p-values wherever possible alongside the summary statistics and 95% confidence intervals. These should be reported for all key questions and not only when the p-value is less than 0.05."

We have added to all figures and/or their corresponding figure legends, including Figure 4C, statistical descriptions for all graphs shown, even if p values are greater than 0.05.

Please note that as we changed the analyses for oPPR as requested by Reviewer 3, we discovered there was a typo in our methods regarding our ephys parameters which should have been currents were elicited between 100 pA and 600 pA (not 200 pA to 600 pA). Secondly, removing the 25 ms interval from oPPR caused the significant interaction between fluid x interstimulus interval to change to non-significant. We have amended the text accordingly to this change in conclusion. As this was not a major part of our overall conclusions, we do not feel this detracts from the study’s primary conclusions.

In addition, in full disclosure, as we analyzed all of our new recording data in females and single drinking session males, we went back over all of our 3-week drinking male data to be absolutely sure every recording was within our allowable parameters (between 100 pA and 600 pA and less that 15% change in resistance). We identified a few cells/technical replicates that were previously missed that were outside our analysis parameters. These have been removed from the prior datasets and new n’s are reported as well as new statistical analyses results where those changes occurred. That being said, there were only a few of these recordings that were identified, and the removal of these recordings did not in any way change any conclusions (i.e. no change between significant/nonsignificant and negligible changes in effect sizes).

7. Please comment in the methods whether reported values include a correction for the liquid junction potential. See other comments section for reviewer 3.

We do not correct for liquid junction potential. We have revised the manuscript to specifically state this (line 701).

8. Please clarify the description of the optogenetic stimulation paradigm in line 213.See other comments section for reviewer 5.

We have provided this clarification (lines 341-350).

Reviewer 1 (Recommendations for the authors):Based on the comments above, these are the recommendations for the authors:1. I strongly suggest increasing the number of mice to have at least 6/7 mice/group for each experiment.

Please see our response in Essential revisions 1.

2. Please report the rationale for starting the ex-vivo recording 24 h after the last binge episode. An alternative interpretation for the reported data could be that the glutamatergic synaptic adaptation observed are simply related to only one (i.e., the last) binge episode and not with the drinking history. Please discuss the possibility that these adaptations can occur with a single binge/alcohol exposure.

Please see our response in Essential revisions 4.

3. In line with the point above, it is not clear why for the electrophysiology recording the control virus is missing. This could be critical also for validating the behavioral approaches.

We appreciate the reviewer’s desire to see the control GFP virus that we used in the behavioral experiments in the ex vivo measures. We must point out though, that as much as we would love to do this experiment it is technically impossible to evoke glutamate release via blue light from solely GFP labeled AIC inputs into the DLS. The data that we present are the best that we can do within the realm of feasibility as far as comparing our in vivo and ex vivo data with each other.

4. Regarding the issue with the unilateral stimulation, ideally the authors should repeat the experiments (at least some of the behavioral controls) to demonstrate similar results. Alternatively, the authors could add a "methodological consideration" section to explain these potential limitations.

This is an interesting point and one that we considered over the course of these experiments. We will explain here our rationale for why we feel this is not a good use of animal life to prove negative data in addition to the high financial cost of doing so. (1) We show that unilateral stimulation of these AIC inputs can control drinking behaviors in specific conditions after plasticity has occurred, but not prior to that plasticity being induced. (2) There is no evidence of any lateralization effect in our data. (3) There are many experiments across the breadth of neuroscience research in which modulating striatal dynamics unilaterally alters behavior, mainly locomotion, which we also show data for being unaltered in control behavioral experiments. The idea that unilateral stimulation is not sufficient to produce a behavioral effect is not supported by our data. (4) If we agree that these synapses are being activated by blue light in vivo in similar ways that they are ex vivo in slice (where we measured the plasticity changes), the takeaway is that the alcohol-induced changes in synaptic responses to optical stimulation do not have an influence on these specific control behaviors – which is the claim we make. (5) Adding a methodological consideration for the in vivo optical stimulation discounts the in vivo effects that we identified that were modulated by unilateral stimulation in our alcohol drinking and lickometry measurements. In the future we can consider performing bilateral stimulation with newer devices that are now available, which we did not have access to when we ran these experiments. In such a study we would specifically test the hypothesis that bilateral activation of AIC input stimulation produces different behavioral responses to unilateral stimulation. Testing that hypothesis, however, is beyond the scope and intent of the present study.

5. I strongly suggest adding female mice to this manuscript to increase the impact of the results.

Please see our response in Essential revisions 2.

6. Please add DAPI to the representative picture of viral injection. It is difficult to appreciate anatomical landmarks for these pictures.

Please see our response in Essential revisions 5.

7. Please keep both introduction and discussion related to the circuit investigation and remove implications for human transition to addiction or the impact of these results on public health.

Please see our response in Essential revisions 3.

Reviewer 2 (Recommendations for the authors):I enjoyed reading this manuscript and have a number of comments for the authors to consider.1. In the introduction, please qualify the criteria of a binge as per NIAAA guidelines, namely 4 drinks in 2 hours for women and 5 drinks in 2 hours for men. Also, there is a recent review on the role of the insula in AUD that updates the cited Barker et al. 2015 review (see Campbell and Lawrence 2021).

Thank you. We have updated our language (lines 34-37) and added this citation (lines 70-75).

2. In the text you state that brain slices are taken 24 hours after the final alcohol session; however, figure 1B implies that slices are taken 48 hours after alcohol. Which is it?

Please see our response in Essential revisions 4.

3. In rodents silencing insula inputs to the accumbens core reduces alcohol self-administration (Jaramillo et al., 2018), and in human heavy drinkers there is increased coupling between the right anterior insula and right nucleus accumbens (Grodin et al., 2018) during high threat alcohol cues. With regards the latter point, in rats the AI mediates relapse-like behavior in a punishment associated context (Campbell et al., 2019). Have the authors performed analogous recordings at AI→accumbens synapses to compare vs the AI→DLS synapses? Also, how much spread of virus was there along the A-P axis of the AI? In this regard rodent studies suggest that functionally inhibiting the caudal portion of the insular cortex reduced alcohol consumption whereas functionally inhibiting the anterior insular cortex. was without effect (Haaranen et al., 2020; Pushparaj and Le Foll, 2015).

These experiments were designed based on our prior work that hinted that AIC inputs to DLS are especially sensitive to the effects of alcohol (Muñoz et al., Nat. Comm. 2018) as noted in the next comment by the reviewer. We have not performed recordings of AIC→accumbens synapses as a comparison, although this would be an interesting avenue for future experimentation. Regarding spread of the viral injections, we show bregma levels for spread in the AIC and discuss selection criteria for exclusion based on hits, we note that more caudal portions of the AIC decrease their density in projection to the DLS. Even so, we are doing all of our work at the terminals where the papers cited are CNO injections where collaterals begin to send projects to the BLA and other brain regions that can conflate projection-specific findings.

4. The authors conclude that "binge drinking alcohol specifically potentiates postsynaptic AMPA and NMDA glutamate receptor responses at AIC→DLS synapses". There is evidence that AI→ventral striatum projections are involved in compulsive food eating (Spierling et al., 2020). As the prior comment, have the authors assessed AI→ventral striatum synapses? Also, while I acknowledge the authors previously showed the disruption of opioid receptor mediated LTD at AIC→DLS synapses was specific to alcohol, and did not occur after sucrose binge (Munoz et al., 2018), have they assessed the specificity of the current data set in relation to alcohol vs other (non-drug or drug) rewards?

Again, we have not assessed AIC→ventral striatum synapses, although we acknowledge this will be an exciting comparison. Considering our prior lack of effect on sucrose consumption (Muñoz et al., 2018), we decided not to explore that here, although we cannot rule out that there may be a difference in these experiments. This is an area we feel is suited for future exploration.

5. Note for discussion, within the DLS in both humans and rodent models, muscarinic M4 receptors are subjected to alcohol-induced adaptations and implicated in alcohol seeking (Walker et al., 2020). Have the authors considered this possibility in terms of binge behavior?

We have specifically focused on glutamate transmission based upon our prior work. We have discussed looking at other neurotransmitter systems in the future as we will eventually work to dissect out the mechanisms underlying the alcohol-induced synaptic adaptations, which could involve M4 receptors or BNDF/p75 as the reviewer suggested here and in their question 9.

6. Figure 2 G,H – why are the data presented as %? Moreover, in figure 2H does alcohol intake increase across weeks in the eGFP group?

Data are percent change within animal to help remove baseline differences in drinking intakes, and to measure solely the effect of AIC input modulation of the DLS on future intake behaviors. We also show the raw data in supplemental figures. Intakes are relatively stable, but obviously they are mice and there will always be an inherent noisiness to the data even at larger N’s. Yes, eGFP increases across weeks in regards to alcohol intake. We did not compute any within group statistics for subsets of data as that was not the point of our study, which was to assess between group differences.

7. Supplemental figure 7B – in the ChR2 group the baseline alcohol intake drops both times prior to the final two binge tests (see sessions 21-30). Does this impact the % change data? Over the same period the baseline intake in the eGFP group is more stable.

We calculate from week 3 and are using sums, so yes they are taken into account. There is likely some cumulative effect of ChR2 in alcohol decreasing by session, but we were not powered to detect it, which is another reason why we chose to measure weekly percent change.

8. Supplemental figure 9 – the drinking profile of the LED off group is unstable, with intake seeming to escalate with time. Is this the case and if so why?

The LED OFF animals were not pretrained to drink from the bottles, which we explain in the methods. We pretrained the LED ON animals on water drinking to ensure any potential decreases with consumption were not related to the inability of animals to drink out of, or neophobia to, the bottles.

9. In terms of a molecular mechanism, have the authors considered investigating the potential role of BDNF / p75 (see Darcq et al., 2016)?

Please see our response to this reviewer’s question 5.

10. Binge drinking in adolescent females is a growing problem, have the authors studied female mice?

Please see our response in Essential revisions 2.

Reviewer 3 (Recommendations for the authors):1) PPR at 25 ms should not be included. The very low values reflect poor fidelity of ChR2 at stimulation frequencies >30 Hz, rather than biological properties of AIC->DLS synapses. Also, the interpretation of these data in line 120 should use specific language to indicate directionality in place of "alter".

We removed 25 ms, which negated the statistical effect. We changed our conclusions accordingly.

2) I suggest switching orders of graphs throughout so that the control groups (Water, eGFP, etc) precede the experimental groups (Alcohol, ChR2, etc)

We thank the reviewer for their style recommendations. Some of the authors agreed with this reviewer’s opinion, but others disagreed. We put it up for a vote in a democratic process and the result of the vote to decide was to keep the style the same. We hope this does not offend the reviewer.

3) I do not understand how the DID intake values for the Water group are tenfold higher than the Alcohol group. Also, "g/kg" is an unusual way to display Water Intake, I think "mL" is more informative and conventional.

The what looks like higher water consumption can likely be accounted for by the fact that the alcohol group is drinking a solution that is only 20% alcohol while the water group is drinking a solution that is 100% water. Once that is accounted for, any discrepancies from what one would expect are likely due to the complex physiological responses to consuming each substance.

While we acknowledge that g/kg of water might be unconventional, to train our machine learning model the two fluids had to have the same units. In addition, the lickometer measures are validated by g/kg as well. In choosing whether to use ml/kg or g/kg for our inputs we felt that g/kg of alcohol was more important to report than ml/kg of water. Our code and thus the math we use to produce these values is publicly available for anyone that wants to convert the data for their own analyses.

4) The representative AMPA/NMDA ratio trace in Figure 1F displays significant AMPA receptor rectification. MSNs do not typically display significant AMPAR rectification under control conditions. Furthermore, the internal solution does not appear to contain spermine, so voltage dependent block of CP-AMPARs should not be a major factor. Did the authors correct for the liquid junction potential?

Thank you for pointing this, we have replaced the traces, because the appearance of what looks like rectification could be due to methodological reasons. We did not correct for liquid junction potential. We have added that point to our methods.

5) The description of the optogenetic stimulation paradigm (line 213) is misleading. As written, it reads as if most of the animals underwent closed-loop stimulation but only a minor portion were open-loop and eventually folded into the main data set. By contrast, the cohort was split n=4/4 open/closed as shown in Figure S6. This section should be re-written accordingly.

We have re-written this for better accuracy to indicate half of the animals experienced the paradigm (see lines 341 – 350).

6) The authors should be commended for assessing the effects of their optogenetic stimulation on long-term plasticity in brain slices. Experiments like this are rarely performed but can provide useful, if not necessary, information for interpreting the effects of in vivo optogenetic stimulation. While no long-term effects were detected, it appears to me as if there could be an Alcohol x Picrotoxin interaction immediately after the 20 Hz stimulation. It seems as if picrotoxin might have decreased the post-tetanus potentiation in slices from Water but not Alcohol mice. If so, this could provide some insight into changes in AIC->DLS feedforward inhibition following DID. Also, these experiments should be described in greater detail in the Methods.

Thank you. We have described this in more detail in the methods and added a comment about these data in relation to feedforward inhibition. We also tested those timepoints (both just the peak right after the stimulation and the total time before the response returns to baseline) and there are no significant effects. Picrotoxin has an effect on oPS amplitude, but there is no interaction with fluid type.

7) While the summarized oPS timecourses are completely blocked by NBQX, the representative oPS traces do not appear to be.

In a further subset of experiments, we applied TTX after NBQX to be sure that NBQX was indeed blocking the entire response. We did not see any further change as a result of TTX application. What may appear to be an incomplete block, is likely part of the stimulus artifact. Our software accounts for this in determining the oPS amplitude. Please see Author response image 1.

**Author response image 1. sa2fig1:** 

8) There seems to be an effect of AIC->DLS stimulation during the first DID cycle in Figure S9.

This is due to methodology and not an effect of stimulation. Please see our response to reviewer 2 point 8 for an explanation.

[Editors' note: further revisions were suggested prior to acceptance, as described below.]

The manuscript has been improved but there are some remaining issues that need to be addressed, as outlined below:1. Amend the title of the manuscript to "The role of anterior insular cortex in binge alcohol drinking" (or similar).

We have updated the title to remove the word govern.

2. Add to the main text that unilateral opto manipulation was used.

We’ve added unilateral to the main text in lines 290 and 348.

3. Include missing citation on Line 1002.

We have rechecked all citations and the reference section should be inclusive of all in text and methods citations.

4. Add the "n" values to the figure 3 legend. Please double-check that all figure legends contain such "n" value information.

We have added this information to the figure 3 legend.

5. Perform power analysis to determine whether the ICSS dataset is adequately powered. State the result (whether or not) concisely in the text. For example (p-value, study sufficiently powered as determined by power analysis).

We performed a power analysis using the data in Figure 4 J to explicitly determine if we were adequately powered to discover a between factor effects for Virus with our current data. To do so, we used the partial eta squared value from the mixed ANOVA to directly compute the Virus effect size in G*Power 3.1. Using that effect size, sample size, number of groups, number of repeated measures, and computing the correlation amongst the repeated measures data, we achieved a power of = 0.9898339, indicating our sample size was correctly powered to discover a main effect of Virus. Also, we were also curious to see if we had the power to discover a within-between interaction and repeated the same approach using the partial eta squared value of the interaction of Virus x Session from our mixed ANOVA analysis to compute effect size, the nonsphericity epsilon value, and the additional values noted above. We achieved a power of = 0.5582789, suggesting although we are powered to discover a main effect for Virus, our ability to discover a Virus x Session interaction is below the threshold of = 0.8.

Thus, we are confident that our claim AIC→DLS photoexcitation does not alter operant responding remains valid given the experimental procedure employed in this manuscript. We note that future analyses may need to address whether stimulation of this pathway over time produces differential effects in operant responding in different contexts. We have added the main effects β value to the Figure 4 legend after the p value for Figure 4J.

6. Add missing citations to the conclusion.

We have added citations and additional text to the conclusion.